# PTSD and PTG in French and American Firefighters: A Comparative Study

**DOI:** 10.3390/ijerph191911973

**Published:** 2022-09-22

**Authors:** Charlotte Henson, Didier Truchot, Amy Canevello

**Affiliations:** 1Laboratory of Psychology, Université Bourgogne Franche-Comté à Besançon, 25000 Besançon, France; 2Department of Psychology, University of North Carolina at Charlotte, Charlotte, NC 28223, USA

**Keywords:** PTSD, trauma, PTG, growth

## Abstract

Studies show that experiencing traumatic events can lead to positive psychological change, or posttraumatic growth (PTG). In the hope of promoting PTG, authors have been focusing on identifying the factors that may foster PTG. Despite these attempts, the literature shows inconsistencies, making it difficult to know which variables may be involved in the process of growth. Indeed, authors seem to disagree on the nature of the relationship between PTSD and PTG, time since the event, social support, intrusive rumination, and sociodemographics. Thus, this study aims to clarify these discrepancies, and verify whether the processes involved are the same across two different cultural groups, both of which are confronted with traumatic events regularly: 409 American firefighters, and 407 French firefighters. Results indicate that, in both samples, PTG is positively related to PTSD, subjective perceptions of the event, stress during the event, disruption of core-beliefs, and deliberate rumination; and unrelated to social support, core-self evaluations, and socio-demographic variables (age, gender, relationship status, etc.). However, time since the event and the number of years on the job only predicted PTG in the American sample, while colleague and emotional support only predicted PTG in the French sample. Additionally, American firefighters reported more growth, more social support, more positive self-perceptions, more intrusive rumination, and more neuroticism than French firefighters. These results suggest that the process of growth, as defined by Tedeschi and Calhoun, is relatively stable among firefighters, but that some differences do exist between cultural groups.

## 1. Background

People who are repeatedly exposed to traumatic events –firefighters, in particular– have a significant risk of developing post-traumatic stress disorder [1,2,3]. In 2015, firefighting was deemed the most stressful profession, due to its unpredictability and negative psychological effects [4]. Additionally, according to Boffa et al. [5], studies have found that anywhere between approximately 7% and 30% of firefighters meet the criteria for a current diagnosis of PTSD. Not only do these rates confirm the dangerous nature of firefighting operations, they also further indicate that firefighters risk developing psychological disorders, such as PTSD [6,7,8], substance abuse [6,9] and depression [10].

However, the literature linking firefighting to trauma shows some inconsistencies. Indeed, while some studies indicate higher rates of PTSD among firefighters (between 18% and 37%;, e.g., [6,7,8], others suggest surprisingly lower rates of PTSD (between 5% and 13%;, e.g., [11,12,13], comparable to those in the general population (6.8% according to the *National Center for PTSD*). While some studies show an increased risk for developing long-term psychological disorders, others suggest high levels of resilience among firefighters and a propensity to heal and recover from hardship [14]. In fact, while studies suggest that nearly all individuals will experience a traumatic event in their life, their reactions to trauma may greatly differ [15]. For some, exposure to a PTE (potentially traumatic event) will trigger the development of pathogenic symptoms, such as symptoms of depression, anxiety and post-traumatic stress [16]. For others, these negative symptoms will give rise to positive psychological benefits. Indeed, for some individuals, the distress that stems from traumatic experiences can be “catalysts” for positive change (i.e., *posttraumatic growth; PTG*). These changes may include “improved relationships, new possibilities for one’s life, a greater appreciation for life, a greater sense of personal strength, and spiritual development” ([17], p. 504).

### 1.1. Relationship between PTSD and PTG

Based on what was previously mentioned, are PTSD and PTG intrinsically related? When looking at the literature, some studies support the positive and significant correlation between these two variables [18,19,20,21,22,23], while others find that PTSD is only correlated with certain domains of growth (e.g., spiritual change; [24]), that there is no correlation whatsoever between the two [25,26,27,28,29,30,31], or that PTSD positively predicts PTG in only certain professional groups (e.g., psychologists; [32]). Furthermore, among the studies that do find a positive association between PTSD and PTG, there are contradictions regarding the nature of the relationship between stress and growth. Indeed, some studies suggest that the trauma must be “seismic” enough (extremely severe) in order to foster growth [33,34,35], while others support the idea that it is “moderate” levels of PTSD that will promote the highest levels of PTG [32,36,37,38]. Indeed, McCaslin et al. [37] explain that “low levels of distress may be insufficient to stimulate growth, and that an overwhelming amount of distress—at the time of the event and following it—may impede the development of growth following traumatic events” (p. 338). Thus, a majority of the studies support the idea that there is a curvilinear relationship between PTSD and PTG [39,40,41,42,43]. Additionally, some authors suggest that it is low subjective stress that will most likely predict growth [29,44,45,46], which is consistent with Martins da Silva et al.’s study [47], suggesting that the event does not even necessarily have to be perceived as “traumatic” in order to trigger PTG. Given these conflicting findings, we tested the relationships between (1) PTSD and PTG, (2) the level of distress experienced at the time of the event and PTG, and (3) the subjective perception of the event (was it perceived as “traumatic”?) and PTG, in both of our samples.

### 1.2. Relationship between Time and PTG

The relationship between the time elapsed since the event [17] and PTG also raises a debate. Indeed, while several authors suggest that posttraumatic growth is a process that requires time [35,48,49], others show in contrast that the time elapsed since the traumatic event is not predictive of growth in samples of breast cancer survivors and veterans with a major combat-related amputation [50,51,52]. However, it is possible to hypothesize that the absence of correlation found in these studies are due to the fact that both types of traumas induce strong *mortality reminders* [53], as the body is heavily affected by these particular events (mastectomy and amputation). Indeed, prominent scars or body lesions may induce a permanent sense of vulnerability or a constant reminder of the endured trauma that could impede the growth process (e.g., having an increased perception of personal strength). Furthermore, Pan et al. [54] show in a sample of bereaved parents who had lost their only child that death by illness was more predictive of PTG than death by accident or suicide. A possible explanation for this could be that illness, in contrast with sudden-death type of events, gives more time for the parents to adjust and get used to the idea that death is a possible outcome. In addition, Taku et al. [55] show that there is in fact a positive correlation between the time since the event and PTG in an American sample, but that no correlation was found between these two variables in a Japanese sample. Similarly, some studies suggest that PTG may increase over time [48], while others suggest that PTG decreases [56], or rather is a dynamic process that may take different trajectories (e.g., increase, decrease, or remain stable; [20,57,58]. Finally, Wu et al. [59] suggest in their literature review that shorter time since the traumatic event was predictive of higher levels of growth. Given these major discrepancies, we tested the relationship between time and PTG within our two large firefighter samples to promote statistical accuracy.

### 1.3. Relationship between Social Support and PTG

The literature also shows that not all authors agree on the functions of social support that promote PTG. Indeed, while several studies suggest that emotional support predicts higher levels of growth [29,60,61,62], others show that emotional support is in fact not correlated with PTG and that instrumental support (or tangible support) is the only social contextual predictor of growth [61,62,63,64]. A possible explanation for this could be that excessive levels of emotional support may trigger feelings of incompetency or lack of self-efficacy, both of which are important predictors of PTG [65,66,67,68,69]. Instrumental support, on the other hand, may signal caring by offering a helping hand for various practicalities (e.g., looking after children or providing food), which may reduce cognitive load for individuals and allow them to focus more on their psychological growth and healing [70]. Additionally, besides emotional and instrumental support, Mairean’s [29] study shows that positive social interactions, informational support [71], tangible support, and affectionate support all correlate positively with posttraumatic growth, but that among these 4 forms of social support, only positive social interactions moderated the effect of traumatic stress on PTG. According to Mairean [29], three explanations are possible: (1) positive social interactions may represent an opportunity for social disclosure in a secure and positive environment; (2) positive social interactions may allow more frequent disclosure; and (3) positive social interactions may promote the use of more effective coping strategies that promote growth. This third point is supported by Zhou et al.’s [72] study, showing that social support directly and indirectly fosters growth through positive cognitive reappraisal. Thus, it appears that one of the most valuable aspects of social support may be that it provides the opportunity to process the traumatic event through interaction, meaning that instrumental support alone may not be enough to foster PTG. Indeed, what may actually matter the most is the amount of support that is being provided: While various forms of social support are necessary, excessive support of any kind may have a counter effect of growth.

Despite the contrasting data, it seems clear that social support, regardless of the type, plays an important role in the development of positive psychological change. In order to clarify some of the contradictions in this literature, we tested the relationship between social support and PTG among our firefighters. Four types of social support were measured: Supervisor support, colleague support, emotional support and professional support. Lastly, because Americans live in a more “masculine” culture than the French –meaning in a society where weakness is not tolerated and men are expected to be assertive, competitive, and focused on material success [73], we expected French firefighters to report higher levels of social support.

### 1.4. Relationship between Rumination and PTG

Additionally, the role of intrusive rumination in the development of posttraumatic growth remains uncertain. Indeed, while some authors suggest a positive association between intrusive rumination and PTG [74,75,76,77,78,79], others find no correlation whatsoever between these two variables (Li et al., 2018), or rather a positive association between intrusive rumination and distress [80]. Similarly, some studies suggest that written assignments, which lead to rumination, will foster growth [81,82], whereas one study shows that writing sessions increase post-traumatic stress and decrease PTG over time [83]. Additionally, some studies show that intrusive rumination only indirectly promote PTG via deliberate rumination [31,72,84]. Given these inconsistencies, we tested the relationships between PTG and intrusive and deliberate rumination in our firefighter samples. By testing the relationship between these variables in two different groups, we hope to increase our chances of clarifying the existing data. Additionally, we tested whether deliberate rumination could be a mediator of the relationship between PTSD and PTG. We then added another variable to our analysis: disruption of core-beliefs. Indeed, studies suggest that an individual’s core-beliefs need to be “severely shaken/shattered” in order to trigger rumination processes and, in turn, elicit the development of growth [35]. Thus, we hypothesized that the disruption of core-beliefs could be a mediator (or promoter) of deliberate rumination and, therefore, a mediator of PTG as well.

### 1.5. Relationship between Personality and PTG

Personality also plays an important role in the experience of PTG. Indeed, studies show that personality traits of the Big Five including agreeableness, extraversion, openness [19,85], and conscientiousness [86,87] are positively correlated with PTG, with openness being the most predictive of positive psychological change [85]. Indeed, given the personality traits associated with openness, e.g., imagination and adventurousness, it is reasonable to hypothesize that individuals high in openness may be more prone to adapt to unexpected life events [88]. In addition to these traits, the sense of control individuals feel they have over themselves and their environment strongly impacts PTG as well. Indeed, individuals who feel they have control over themselves or the situation they are facing usually feel more capable of managing problems [89]. Additionally, studies show that self-control (or personal mastery) is, on the one hand, a protective factor against PTSD [90], and on the other hand, a key determinant of psychological adjustment after adversity [91,92]. Given the large number of studies that assess the relationship between PTG and the personality traits of the Big Five, we wanted to test the link between growth and a different aspect of personality: Core self-evaluations (CSE), which have not been studied in relation to PTG so far. CSEs are a stable personality trait defined by people’s fundamental evaluations about themselves, their abilities and their personal control. Individuals who score high in CSE tend to be confident in their own abilities and have more positive views about themselves, while individuals who score low in CSE lack confidence and view themselves in a more negative light. Thus, we hypothesized that firefighters who scored high in CSE would report lower levels of PTSD and higher levels of PTG than firefighters with lower CSE. Additionally, we hypothesized that participants higher in neuroticism would report lower levels of growth. Lastly, as authors have demonstrated that sense of control, self-efficacy, and self-worth [65,66,67,68,69] are important predictors of PTG, we expected that locus of control, self-esteem, and self-efficacy positively predict PTG.

### 1.6. Relationship between Sociodemograohics and PTG

Lastly, according to several studies, sociodemographic characteristics such as gender, age, and relationship status also play an important role in the onset of posttraumatic growth. Thus, we decided to assess the relationships between these sociodemographic variables and PTG in our firefighter samples. More precisely, we tested the relationship between PTG and gender, age, number of years on the job, rank, relationship status, whether participants had kids or not, and kids’ age. We first hypothesized that female firefighters would report higher levels of PTG than male firefighters, as it is often found in the literature that women report higher levels of growth than men, specifically on the “relating to others” dimension [45,93,94,95,96]. With regard to the relationship between PTG and age, studies again show contradictions. Indeed, some studies find that younger people tend to report higher levels of growth than older people [95], while others support the idea that older people experience the highest levels of PTG [97,98]. However, for firefighters in particular, we hypothesized that older participants may report higher levels of growth, as perhaps the more experience firefighters have, the better they are able to cope with trauma and develop positive psychological change. We then hypothesized that ranking may impact the onset of PTG. Indeed, the higher the rank in the fire department, the higher the levels of responsibility. Therefore, it could be expected that firefighters with higher ranking may experience more severe levels of stress which, in turn, could make them more susceptible to PTSD. We then tested the link between PTG and firefighters’ relationship status. We expected these two variables to be strongly related, as the literature shows that supportive relationships are indispensable for individuals’ experience of growth in the aftermath of trauma [29]. Lastly, we hypothesized that having children may impact the onset of PTSD and PTG. Indeed, the literature shows that being confronted with sick, injured or dying children may be one of the most stressful events for health and emergency professionals [29]. Thus, by a process of projective identification, we expected stress levels to be worse for firefighters who are also parents. Additionally, we imagined that these types of events would be worse for parents of younger children, as they already tend to be more worried and mindful of their child’s well-being and safety.

Lastly, we examined whether the growth process remained the same across different cultures. More specifically, we decided to test which variables would affect the onset of PTSD and PTG in one American group, and one French group. Based on Hofstede’s *dimensional model of culture* [73], we can define “culture” as the “collective mental programming of the human mind which distinguishes one group of people from another” [73]. Hofstede describes countries based on 6 different dimensions: Masculinity, uncertainty avoidance, indulgence, power distance, individualism, and long-term orientation. These dimensions strongly influence patterns of thinking and behaviors. As Americans score differently than the French on all of these dimensions [99], it is indeed possible to expect some differences between samples of European and North American firefighters. In this study, and in both samples, we test the strength of the relationship between PTG and PTSD, the time elapsed since the event, social support, sociodemographic characteristics, personality, and rumination, with the hopes of clarifying most of the divergences that were put forward. Thus, this study aims to (1) offer a better intercultural understanding of post-traumatic stress and growth in first responders, particularly firefighters, and (2) test the robustness of the process of PTG.

## 2. Method

### 2.1. Participants

Four hundred and nine American firefighters and 406 French firefighters participated in this study, for a total of 815 participants. American firefighters were recruited in 28 different fire stations in Charlotte, North Carolina. French firefighters were recruited in 14 fire stations in France in the departments of Doubs and Belfort. Female firefighters represented 2.7% of all American participants and 13.2% of all French participants.

In the **American sample**, participants ranged in age from 20 to 59 (M = 37; SD = 8.4). Two hundred sixty-nine reported having children. Children’s ages ranged between 0 (unborn) and 32. Participants’ number of years on the job ranged from 1 to 45 years (M = 15.3; SD = 8.2). Three hundred seven were Probationary Firefighters, Firefighter/Paramedic, or Driver Engineer (responsible for the hands-on actions of fire suppression and search and rescue), 94 were Lieutenant or Captain (responsible for managing operations on the scene of an emergency), and 8 were Battalion Chief, Assistant Chief, or Fire Chief (responsible for the efficient operation of the fire department).

In the **French sample**, participants ranged in age from 18 to 64 (M = 37; SD = 10.7). 270 reported having children. Children’s ages ranged between 0 (unborn) and 33. Participants’ number of years on the job ranged from 1 to 49 years (M = 18; SD = 10.1). 184 were “Hommes du rang” (similar to Probationary Firefighters, Firefighter/Paramedic, or Driver Engineer), 187 were “Sous-officiers” (similar to Lieutenant or Captain), and 29 were “Officiers” (similar to Battalion Chief, Assistant Chief, or Fire Chief).

### 2.2. Informed Consent

Before being administered the survey, participants were asked to read our consent document which requires them to check a box indicating they have read and agree with the terms and conditions. The consent form included information about the objectives of this study, the length of the survey, the protection of their personal information, their rights as volunteer participants, and contact information about the researchers. If a participant did not consent, they simply did not answer the survey.

### 2.3. Measures

Participants completed a 20-min paper survey. Participants first completed a self-report measure designed to screen for potentially traumatic events. They were then asked to think about the most stressful event they experienced within the framework of their profession, to describe it, and to answer the rest of the questions with respect to that event. Participants also reported how long ago the event occurred, and completed measures of post-traumatic stress, disruption of core beliefs, post-traumatic growth, intrusive and deliberate rumination, social support at work, and core-self evaluations. Finally, they reported demographic information.

**Potentially traumatic events** were screened using the *Life Events Checklist for DSM-5* (LEC-5; [100]). Participants were presented with a list of 17 stressful events, and were asked to indicate whether: (a) it happened to them personally; (b) they witnessed it happen to someone else; (c) they learned about it happening to a close family member or close friend; (d) they were exposed to it as part of their job; (e) they’re not sure if it fits; or (f) it does not apply to them. Participants were asked to consider their entire life (growing up as well as adulthood) as they went through the list of events. The LEC-5 is intended to gather information about the potentially traumatic experiences a person has experienced; thus, it does not yield a total score or composite score. This scale was used to prime participants to think about the various stressful events they may have encountered in their life, before asking them to describe the most distressful one they’ve experienced within the framework of their profession.

**Post-traumatic stress** was assessed using the 17-item *PTSD Checklist for DSM-5* (PCL-C; [101]). Participants were asked to think about the most stressful event they experienced within the framework of their profession, and to then read each problem listed in the PCL-C and indicate how much they had been bothered by the problem in the past month (e.g., “Repeated, disturbing dreams of the stressful experience”; “Feeling distant or cut off from other people”; “Feeling irritable or having angry outbursts”). All items were rated on a scale ranging from 1 (*not at all*) to 5 (*extremely*).

**Disruption of Core beliefs** were measured using the 9-item Core Beliefs Inventory described by Cann and colleagues [102]. The items focus on religious and spiritual beliefs, human nature, relationships with other people, meaning of life, and personal strengths and weaknesses. The instructions indicated that participants should reflect upon the event about which they were reporting and indicate the extent to which it led them to seriously examine each core belief (e.g., “The degrees to which I believe things that happen to people are fair.”). Items were rated on a scale from 1 (*not at all*) to 5 (*very much*).

**Posttraumatic Growth** was assessed using the 25-item *Posttraumatic Growth Inventory* (PTGI; [103]). Participants were shown 25 statements describing positive change and asked to indicate for each statement the degree to which each had occurred in their life as a result of the stressful event they identified earlier. Seven items measured relating to others (e.g., “I have a greater sense of closeness with others”); five items measured new possibilities (e.g., “I established a new path for my life”); four items measured personal strength (e.g., “I know better that I can handle difficulties”); six items measured spiritual change (e.g., “I have a better understanding of spiritual matters”); and three items measured appreciation of life (e.g., “I have a greater appreciation for the value of my own life”). All items were rated on a scale ranging from 0 (*I did not experience this change as a result of my crisis*) to 5 (*I experienced this change to a very great degree as a result of my crisis*).

**Intrusive and Deliberate Event-related rumination** was measured using the 20-item Event-Related Rumination Inventory described by Cann and colleagues [104]. The first 10 items assessed intrusive rumination, and began with the stem “*After an experience like the one you reported, people sometimes, but not always, find themselves having thoughts about their experience even though they don’t try to think about it. Indicate for the following items how often, if at all, you had the experiences described during the weeks immediately after the event.*” (e.g., “Thoughts about the event came to my mind and I could not stop thinking about them”). The 10 following items assessed deliberate rumination, and began with the stem “*After an experience like the one you reported, people sometimes, but not always, deliberately and intentionally spend time thinking about their experience. Indicate for the following items how often, if at all, you deliberately spent time thinking about the issues indicated during the weeks immediately after the event.*” (e.g., “I forced myself to deal with my feelings about the event”). All items were rated on a scale from 1 (*not at all*) to 5 (*extremely*). Global rumination scores (both combined) and scores for each specific type of rumination were calculated.

**Social support at work** was measured using the 8-item Karasek social support scale [105]. Four of the items measured social support by the supervisor (e.g., “My supervisor helps me successfully complete my tasks”), while four others measured social support by colleagues (e.g., “The colleagues with whom I work show interest in me”). Among these eight items, four focused on professional support (e.g., “The colleagues with whom I work are professionally skilled”), while four focused on emotional support (e.g., “My supervisor pays attention to what I say”). All items were rated on a scale from 1 (*strongly disagree*) to 4 (*strongly agree*).

**Core Self-Evaluation** was measured using the 12-item Core-Self Evaluation Scale (CSES; [106]). The CSES assesses 4 core self-evaluation traits: (1) *Self-esteem*, or the overall value that one places on oneself as a person, e.g., “Overall, I am satisfied with myself” [107]; (2) *generalized self-efficacy*, or an evaluation of how well one can perform across a variety of situations, e.g., “When I try, I generally succeed” [108]; (3) *neuroticism*, or the tendency to have a negativistic cognitive/explanatory style and to focus on negative aspects of the self, e.g., “Sometimes when I fail, I feel worthless” [109]; and (4) *locus of control*, or beliefs about the causes of events in one’s life, e.g., “I determine what will happen in my life” (internal vs. external causes; [110]). The CSES uses a 5-point Likert scale ranging from 1 (*strongly disagree*) to 5 (*strongly agree*).

## 3. Results

### 3.1. Relationship between the Time of the Event and PTG

In order to test the relationship between time and PTSD within both of our firefighter samples, we calculated bivariate Pearson correlations. Our results indicate that time was unrelated to post-traumatic stress in both groups (r = −0.03 for the French and r = 0.04 for the US; see Table 1). However, time since the event was unrelated to PTG in our French sample (r = 0.029), but significantly correlated with PTG in our American sample (r = 0.12, *p* < 0.05). Nevertheless, the strength of this positive correlation remained weak (r² = 1%).

### 3.2. Relationship between PTG and PTSD

An independent samples *t*-test first revealed that there were no significant differences in PTSD levels between French and American firefighters (t (802) = −1.161; NS). However, there was a significant difference in PTG levels between our two groups (t (773) = −3.77; *p* < 0.05). Indeed, it appears that American participants report higher levels of growth than French participants (see all mean comparisons in Table 2). When looking at the different subscales of the PTGI, other *t*-tests showed that Americans specifically score higher on relating to others (t (798) = −4.44; *p* < 0.001), spiritual development (t (796) = −5.29; *p* < 0.001), appreciation of life (t (804) = −3.59; *p* < 0.001), and personal strength (t (806) = −2.58; *p* < 0.001). No significant differences were found on the new possibilities subscale of the PTGI (t (804) = −1.05; NS).

Results then indicate that the PCL-C (PTSD) and PTGI were positively and significantly related in the French (r = 0.43, *p* < 0.01) and American samples (r = 0.35, *p* < 0.01; see all correlations in Table 1). In other words, post-traumatic stress was positively associated with growth. Furthermore, the level of stress experienced at the time of the event was positively and significantly related to both PTSD and PTG among French (r = 0.34, *p* < 0.01 and r = 0.34, *p* < 0.01, respectively) and American firefighters (for both, r = 0.22, *p* < 0.01). Thus, the more distressed firefighters felt at the time of the event, the more post-traumatic stress and posttraumatic growth they experienced afterwards. Finally, our data show that the subjective perception of the event (e.g., the individual’s personal experience, interpretation, and feelings about the event), was also positively related to PTSD (r = 0.29, *p* < 0.01 in the French sample and r = 0.15, *p* < 0.01 in the American sample) and posttraumatic growth (r = 0.25, *p* < 0.01 in the French sample and r = 0.11, *p* < 0.01 in the American sample), which means that participants reported higher levels of PTSD and PTG when the event was perceived as “traumatic”.

### 3.3. Relationship between PTG and Social Support

We tested the relationships between PTG and four types of social support using the Karasek Social Support Scale: supervisor support, colleague support, emotional support, and professional support. An independent samples *t*-test first revealed that there was a significant difference in total scores on the Karasek Scale between our French and American samples (t (804) = −15.53; *p* < 0.05). Indeed, French firefighters appear to report lower levels of social support than American firefighters. Similarly, another t-test revealed significant differences in scores on all 4 types of social support measured by the scale. More specifically, French firefighters reported lower levels of supervisor support (t (808) = −13.58; *p* < 0.001), colleague support (t (805) = −13.00; *p* < 0.001) emotional support (t (805) = −15.14; *p* < 0.001), and professional support (t (808) = −13.81; *p* < 0.001) than Americans. Furthermore, results indicate that supervisor support and professional support were not correlated with PTG in the French (r = 0.037; *p* > 0.05 and r = 0.040; *p* > 0.05, respectively) and the American sample (r = 0.017; *p* > 0.05 and r = −0.064, *p* > 0.05, respectively). However, colleague support and emotional support were both correlated with PTG in our French sample (for both, r = 0.10, *p* < 0.05), but unrelated to PTG in our American sample (r = −0.060, *p* > 0.05 and r = 0.023, *p* > 0.05, respectively). Finally, global support (all four types combined) was not significantly correlated with growth in either of our groups (r = 0.078 for the French and r = −0.018 for the US).

### 3.4. Relationship between PTG and Personality (The Core-Self Evaluations Scale)

An independent samples t-test first showed that there was a significant difference in CSE scores between French and American firefighters (t (793) = −10.4; *p* < 0.05). Indeed, the American sample scored higher on the CSE scale than the French sample. More specifically, the US sample scored higher on all 4 subscales of the CSE: Self-esteem (t (805) = −7.43; *p* < 0.001), self-efficacy (t (803) = −4.14; *p* < 0.001), neuroticism (t (806) = −7.83; *p* < 0.001), and locus of control (t (807) = −12.22; *p* < 0.001). Additionally, PTG was unrelated to the CSE total score in both groups (r = −0.070; NS in the French sample; r = −0.090; NS in the US sample). However, PTSD was negatively but significantly related to the CSE scale (r = −0.22, *p* < 0.01 in the French sample; r = −0.30, *p* < 0.01 in the US sample). In other words, high levels of PTSD were associated with lower scores on the CSE. Lastly, we tested the relationship between PTG and each of the 4 subscales of the CSE scale (Self-esteem, Self-efficacy, Neuroticism, and Locus of control). Analyses show that, in both samples, the only subscale that was negatively but significantly related to PTG was Neuroticism (r = −0.20, *p* < 0.01 in the French sample and r = −0.15, *p* < 0.01 in the American sample). All three of the other subscales were not significantly related to growth (see Table 1).

### 3.5. Relationship between PTG and Rumination/Core-Beliefs

We first note that no significant differences were found in global rumination scores between French and American firefighters (t (791) = −1.53; NS). However, US firefighters (M = 21.5, SD = 11) scored significantly higher on intrusive rumination than French firefighters (M = 19.7, SD = 8.7; t (796) = −2.60; *p* < 0.05). No significant differences were noted on deliberate rumination (t (806) = 0.42; NS). Results then show that, as expected, rumination was positively and significantly related to PTSD in both groups (r = 0.56, *p* < 0.01 in the French sample; r = 0.51, *p* < 0.01 in the American sample). We also note that both intrusive and deliberate rumination were positively and significantly related to PTG. More specifically, intrusive rumination was positively correlated with PTG at r = 0.30 in our French sample and r = 0.29 in our American sample (both *p* < 0.01), while deliberate rumination was positively correlated with PTG at r = 0.55 in our French sample and r = 0.49 in our American sample (both *p* < 0.01).

However, as intrusive and deliberate rumination are significantly related in both groups (r = 0.55, *p* < 0.01 in the French sample; r = 0.57, *p* < 0.01 in the American sample), it is possible to hypothesize that intrusive rumination may mostly promote growth via deliberate rumination, as suggested by previous authors [84]. We tested this hypothesis with linear regressions, which indicate that, while intrusive rumination and PTG are positively and significantly related, once deliberate rumination is added into the equation, the effect of intrusive rumination on growth disappears in both the French (β = 0.007, NS) and the American sample (β = 0.024, NS). However, the effect of deliberate rumination remained significant in both groups (β(French) = 0.55, *p* < 0.001; β(American) = 0.47, *p* < 0.001). Then, we tested whether the relationship between PTSD and PTG could be mediated by deliberate rumination. Linear regressions show that, once deliberate rumination is included in the equation, the strength of the relationship between PTSD and PTG decreases significantly (β(French) = 0.21, *p* < 0.001; β(American) = 0.20, *p* < 0.001). Lastly, results indicate that, once core-beliefs are added into the equation, the strength of the relationship between PTSD and PTG remains significant but strongly decreases (β(French) = 0.18, *p* < 0.001; β(American) = 0.21, *p* < 0.001). Additionally, once core-beliefs are added into the equation, the strength of the relationship between PTSD and rumination remains significant but strongly decreases as well (β(French) = 0.31, *p* < 0.001; β(American) = 0.25, *p* < 0.001).

### 3.6. Relationship between PTG and Sociodemographics

An independent samples t-Test first showed that there were no significant differences in PTSD and PTG scores between men and women, in either the French (t (395) = 0.36; NS; t (381) = 0.34; NS, respectively) or the American sample (t (402) = 0.47; NS; t (387) = 0.49; NS, respectively). Furthermore, another t-test showed that there were no significant differences between men and women on the “Relating to others” subscale of the PTGI, in either the French (t (391) = −0.69; *p* > 0.05) or the American group (t (402) = 0.72; *p* > 0.05). Next, a bivariate Pearson Correlation showed that age was unrelated to PTSD and PTG, in both French (r = 0.050 and r = 0.031, respectively) and American firefighters (r = 0.040 and r = 0.083, respectively). Similarly, the number of years on the job was not significantly correlated to PTSD among French (r = 0.028) and American participants (r = 0.076). However, analyses show that the number of years on the job was unrelated to PTG in our French sample (r = 0.049), but positively and significantly related to PTG in our American sample (r = 0.11; *p* < 0.05); which means that the more experience American firefighters had, the more growth they experienced. Additionally, a one-way ANOVA showed that no significant differences were found in PTSD scores between our different rank groups (FR: F (2, 397) = 0.98; NS; US: F (2, 404) =0.10; NS). Rank had no significant impact on PTG scores either (FR: F (2, 383) = 1.04; NS; US: F (2, 389) = 2.32; NS). Similarly, relationship status did not have a significant impact on PTSD and PTG, in either our French (F (5, 397) = 0.32; NS; F (5, 383) = 1.24; NS, respectively) or American sample (F (5, 403) = 0.94; NS; F (5, 388) = 0.46; NS, respectively). Lastly, no significant differences were found in PTSD and PTG scores between firefighters who had children and those who did not, in either the French (t (384) = 0.18; NS; t (398) = 0.63; NS, respectively) or the American group (t (387) = −0.33; NS; t (402) = −0.34; NS, respectively). Additionally, kids’ age was not significantly correlated with PTSD and PTG, in either French (r = 0.041; NS and r = 0.025; NS, respectively) or American firefighters (r = 0.070; NS and r = −0.089; NS, respectively).

## 4. Conclusions

The aim of this study was to test the robustness of the process of PTG in firefighters from two different cultural groups: One American group, and one French group. Indeed, by testing whether the growth process remains consistent across different cultures, we demonstrate the reliability and validity of the model.

Our study showed that post-traumatic stress was positively associated with the development of positive psychological change in the aftermath of trauma, which is consistent with a large number of studies [20,21,23] and confirms our initial hypothesis. However, we do note that American firefighters scored significantly higher on PTG than French firefighters. More specifically, they scored higher on 4 of the subscales of the PTGI: Spiritual development, relating to others, appreciation of life, and personal strength. These differences could be explained by the fact that US firefighters may be more religious than French firefighters. Indeed, the literature shows that Americans report higher and more stable church attendance than European countries [111]. Thus, our US sample may use religious coping as their primary coping mechanism when facing adversity, which explains their higher score on spiritual development. Additionally, studies suggest that religious coping could have various positive consequences for individuals in their daily lives, such as giving meaning to negative events, providing a sense of control and comfort during difficult times, fostering social relations through the religious community, and helping individuals make major life changes [15]. Thus, being involved with religion could also explain why Americans scored higher on relating to others, personal strength, and appreciation of life: By being a part of a close-knit community, US firefighters may feel more connected to others. Additionally, by providing a sense of control, religion may help individuals feel more confident in the face of adversity which, consequently, will promote feelings of personal strength. Lastly, as religiosity helps give meaning to life experiences, it may also promote the appreciation of life for what it is, despite its various challenges. Moreover, we note that no significant differences were reported in PTSD scores between French and American firefighters. In other words, despite the fact that both groups reported similar levels of distress, one group still reported higher levels of growth. This observation further emphasizes the fact that PTG is not only determined by stress levels, and that cultural background is an important variable to take into consideration when working with trauma-exposed individuals.

In regard to sociodemographic variables, no significant differences were reported in PTG scores between male and female firefighters, in either of our samples. This finding largely contradicts what the literature has been suggesting for years: That women score higher on PTG than men [45,94,95]. We could explain this by referring to firefighters’ organizational culture (the pattern of shared values, beliefs, and assumptions considered to be the appropriate way to think and act in a given organization), in which “masculinity” plays a major role. Masculinity can be described as the extent to which individuals are motivated by competition, achievement, heroism, assertiveness, and success, above caring for others and quality of life [73]. Given this definition, it is possible to hypothesize that all firefighters, whether male or female, necessarily possess “masculine traits” (as defined by Hofstede), as the very nature of their profession is to accomplish heroic deeds. These traits are reinforced by slogans such as “sauver ou périr” (*save or perish*), “courage et dévouement” (*courage and devotion*), or “brotherhood, dedication, devotion, and service”. Thus, as masculine traits are shared and present among both male and female firefighters, it is understandable that PTG may not be influenced by gender in this specific population. Nevertheless, we note that the lack of relationship between PTG and gender could also be due to the fact that the number of female participants in this study was too small to be truly significant.

Then, age was not correlated with PTSD or PTG in either of our groups. In other words, whether participants were in their 20′s or their 50′s, they obtained similar scores on each of the scales. A possible explanation could be that, no matter how old they are, firefighters are exposed to the exact same challenges on a regular basis. Indeed, recruiting requirements to become a firefighter include being “at least 18 years old” (see www.joingrfire.com, accessed on 19 September 2022). Young recruits are immediately trained for fire protection, medical response, extrication, hazmat response, and water and tech rescue (www.joingrfire.com, accessed on 19 September 2022). Thus, young firefighters, in contrast to young civilians, may mature more rapidly and, therefore, report similar levels of stress and growth than older individuals who have already acquired a large panel of life experiences. Similarly, rank was not correlated to PSTD or PTG in either of our groups. The explanation could be similar to the previous one: Firefighters are confronted with the exact same traumatic scenes, no matter the braid that’s on their uniform. Thus, whether firefighters figure at the top or the bottom of the hierarchical scale, trauma does not discriminate and is above all perceived through individuals’ subjective lens [112]. Additionally, the number of years on the job was not correlated to PTSD in either group, but was however positively and significantly correlated to PTG in the American group only. Similarly, time since the event and PTG were only related in the American group as well, which confirms Taku et al.’s [55] study showing a positive association between time and PTG in their American sample. Thus, it seems that US firefighters may cognitively process traumatic events over time, while the processing of these events may depend on other variables for French firefighters. Lastly, we expected parents, especially parents of young children, to experience more stress and growth, perhaps because they’d be more prone to identifying with situations that involved children, as these were recognized as the most stressful for firefighters [29]. However, having children, no matter their age, was unrelated to PTG in both groups. A possible hypothesis for this could be that firefighters may quickly acquire the capacity to separate their personal life from their professional one. Indeed, by letting themselves be too affected by the emergency calls they respond to, it would be impossible for them to keep working efficiently in the fire department. We could further hypothesize that the uniform plays a major role in this ability. Indeed, uniforms play a huge part in the sense of belonging, and conveys strong group identity [113]. Thus, by putting it on as soon as they enter the fire house and taking it off as soon as they leave, firefighters may use their uniform as a way to mark the barrier between the inside and the outside.

In regard to personality, results show that American firefighters scored higher on the Core self-evaluations scale than French firefighters. More precisely, Americans scored higher on self-esteem, self-efficacy, and locus of control; which means that Americans tend to view themselves as having more personal value, being able to perform well in a variety of situations, and having more control over their environment than the French. The reason why Americans score higher on the locus of control subscale could be explained by referring to the “uncertainty avoidance” dimension of Hofstede’s *dimensional model of culture*. Indeed, Hofstede defines uncertainty avoidance as the extent to which an individual feels the need to control their environment. According to the author, Americans score lower on uncertainty avoidance than the French, which means that Americans do not feel the need to “control” their environment as much and may be more open to change and innovation. Thus, by not having this need, Americans may not feel like their environment is as “uncontrollable” as the French, which could give them the impression they exercise more control over it. Additionally, the reason why Americans score higher on self-esteem and self-efficacy could be explained by referring to Hofstede’s “indulgence” dimension. Countries with a high indulgence score usually allow and encourage free gratification of people’s drives and emotions, while countries with a low indulgence score put more emphasis on suppressing gratification and imposing stricter social norms that regulate people’s behaviors. As the French score lower than the Americans on indulgence, we could hypothesize that, by living in a more restrictive society, French firefighters may have less room for feelings of self-fulfillment and personal achievement which, consequently, could result in less self-esteem and self-efficacy. Lastly, and inconsistently with these results, Americans also scored higher on the neuroticism subscale of the CSE, which represents the tendency to experience negative emotions such as anger, depression, anxiety, and helplessness. Thus, while American firefighters view themselves more positively than the French, they also struggle with more negative affect. A possible explanation could be that, as explained by Hofstede, countries who score high on uncertainty avoidance (such as France) allow for more emotional outbursts and expressions of anger than countries who score low on this dimension. Thus, we could hypothesize that, as the French may express these negative feelings more often, they are able to vent and externalize more of their inner struggles than Americans. Then, statistical analysis showed that the Core self-evaluations scale was negatively but significantly related to PTSD in both the French and the American sample, which means that having positive views about oneself constitutes a protective factor against post-traumatic stress [114]. Additionally, results also revealed that total scores on the CSE were unrelated to PTG. However, when looking at the different subscales of the CSE, neuroticism was significantly correlated to growth and predicted lower levels of PTG in both samples. The other 3 subscales of the CSE were unrelated to PTG, which contradicts the literature suggesting that the sense of control, general self-efficacy, and self-worth are predictors of growth. Finally, despite the fact that Americans score higher on neuroticism, they also report higher levels of growth than the French, which contradicts our initial hypothesis. Thus, our results suggest that personality may not be predictive of PTG in all populations, and that other more prominent variables may come into play in firefighters’ experiences of growth.

Our results show that there were no significant differences in total rumination scores between French and American firefighters; however, Americans scored slightly higher on intrusive rumination. A possible explanation could be that, since French culture allows for more expressions of unpleasant feelings (see Hofstede’s dimension of “uncertainty avoidance”), French firefighters may have the opportunity to vent negative emotions more frequently, which may reduce the appearance of intrusive thoughts. Then, analysis showed that deliberate rumination was strongly correlated to PTG in both samples, and that, once deliberate rumination was added into the equation, the strength of the relationship between PTG and PTSD decreased significantly. Thus, while PTG is partly explained by PTSD, deliberate rumination plays a strong role in the development of positive psychological change, making deliberate rumination a partial mediator of growth. Additionally, analysis showed that core-beliefs partially mediated the relationship between (1) PTSD and deliberate rumination, and (2) PTSD and PTG. Thus, while PTSD and PTG are strongly related, it becomes obvious that the cognitive processes that follow PTSD (disruption of core-beliefs & rumination) are essential for growth to occur. Lastly, intrusive rumination was only associated to growth when deliberate rumination was included in the equation. Thus, contrarily to what the literature might suggest, intrusive rumination may only promote the development of growth by encouraging individuals to think deliberately about their traumatic event.

Lastly, despite the fact that nearly all studies show that social support plays a major role on stress and growth in the aftermath of trauma, it appears that global social support (all 4 subscales combined) does not predict PTG in firefighters, regardless of cultural background. However, the “colleague support” and “emotional support” subscales of the Karasek scale were positively correlated to PTG in the French sample, but completely unrelated to PTG in the American sample. This could be explained by referring once more to Hofstede’s “uncertainty avoidance” dimension: As stated previously, since the French express negative emotions more often, they may benefit more from the emotional support of their colleagues. However, we do note that this correlation was weak and only existent in one of our groups. A possible explanation could be the harmful effects of “toxic masculinity”, in other words, the set of standards our society holds for men, combined with the standards firefighters set for themselves as “soldats du feu” (in English, “fire soldiers”). Indeed, men, and firefighters in particular, are expected to be strong, unafraid, and to have no feelings [115]. These expectations are further accentuated by the famous mottos firefighters live by, such as “Courage et dévouement” (*Courage and dedication*) and “Sauver ou périr” (*Save or perish*). These unrealistic expectations inevitably shape firefighters’ behaviors, especially towards help or support seeking. Thus, the lack of relationship between global social support and PTG in our samples could be explained by the fact that this professional field does not encourage firefighters (men and women) to seek for help and support. This hypothesis is further confirmed by the lack of correlation between relationship status and PTG. Indeed, whether participants were in a romantic relationship, divorced, single, or widowed, they showed similar levels of PTSD and PTG, which suggests that having a partner that could provide social support does not influence growth. Therefore, firefighters may turn to other coping strategies when facing adversity. However, if social support were to be encouraged, it seems that peer support and emotional support would be the most predictive of growth among firefighters, as suggested by the positive association between these forms of support and growth in the French sample. Lastly, analysis showed that French firefighters reported significantly lower levels of social support than American firefighters on all 4 subscales of the Karasek support scale (emotional support, professional support, colleague support, supervisor support). This contradicts our initial hypothesis, as we expected US firefighters to seek less for help given their higher score on Hofstede’s “masculinity” dimension. A possible explanation could be that, since Americans are used to evolving in a society where weaknesses are suppressed among men [23], they may not perceive the absence of social support as strongly as French firefighters. Indeed, we could hypothesize that individuals perceive the lack of social support when they actually expect to receive it. Another hypothesis could be that Americans may utilize other forms of social support than the ones that were assessed in this study. In sum, our data suggests that the way social support is perceived and utilized may greatly depend on cultural background.

All in all, despite obvious cultural differences between French and Americans, it seems that the processes that illicit the development of growth remain relatively stable among firefighter populations. Therefore, this means that PTG is a robust concept that stays consistent and valid among individuals, regardless of their cultural background. Indeed, in both groups, PTG was significantly predicted by PTSD, stress experienced during the event, subjective perceptions of the event, deliberate rumination, and the disruption of core-beliefs. This largely confirms Tedeschi and Calhoun’s theory [35] stating that, in order to experience growth, individuals must feel like their world has been severely challenged, and must cognitively engage to make sense of the event. However, some differences were reported: Time since the event and the number of years on the job only predicted PTG in the American sample, while colleague support and emotional support only predicted PTG in the French sample. Additionally, American firefighters reported more posttraumatic growth, more social support, more positive self-perceptions (self-esteem, self-efficacy), more intrusive rumination, and more negative affect (neuroticism) than French firefighters. These findings can strongly benefit the field of psychology, as they offer a better understanding of the similarities and divergences that exist among firefighter populations, depending on their country of residence. Additionally, this study clarified some of the major contradictions that were identified in the literature (e.g., the role of intrusive rumination). We sincerely hope this new data offers new tools for psychologists, especially those working alongside trauma victims and emergency personnel.

## 5. Limitations

Despite the positive contributions of this study, we note that there are limitations to this research. More particularly, social-desirability bias, or the tendency to behave or respond to questions in a way that will be viewed favorably by others [116], could be a concern for this type of study. Indeed, it is possible that some participants under-reported what they might consider “bad” responses in our surveys, as post-traumatic stress is still stigmatized in this masculine-prominent field of work. Thus, we suggest that future research combine the use of the PCL-C and the PTGI (which are self-report based) with the evaluation of a clinical practitioner.

Additionally, we did not compare French firefighters’ work to American firefighters. Indeed, in some countries, firefighters are only responsible for fighting fires, whereas in others, firefighters have to respond to a much diverse range of calls, including medical and social assistance. The differences that may exist between French and American firefighters’ activities could have impacted our results. We highly suggest future studies evaluate those differences and their potential influence on the development of positive psychological change.

## Figures and Tables

**Table 1 ijerph-19-11973-t001:** PTSD, PTG, core-beliefs, rumination, social support and core self-evaluations: Correlations analysis in French (*n* = 406, lower values) and American firefighters (*n* = 409, upper values).

	Means	SD	1	2	3	4	5	6	7	8	9	10	11	12	13	14	15	16
PTSD	23.69	7.57	1	0.35 **	0.37 **	0.51 **	0.52 **	0.36 **	−0.10 *	−0.11 *	−0.080	−0.10 *	−0.088	−0.30 **	−0.30 **	0.046	−0.40 **	−0.19 **
2. PTG	50.00	21.73	0.43 **	1	0.45 **	0.43 **	0.30 **	0.49 **	−0.018	0.017	−0.060	0.023	−0.064	−0.090	−0.059	−0.002	−0.15 **	−0.044
3. Core-beliefs	17.08	6.73	0.46 **	0.60 **	1	0.40 **	0.31 **	0.40 **	−0.034	−0.024	−0.042	−0.034	−0.034	−0.18 **	−0.19 **	0.026	−0.23 **	−0.12 *
4. Rumination (total)	38.14	14.02	0.48 **	0.56 **	0.49 **	1	0.92 **	0.85 **	0.032	0.018	0.040	0.040	0.019	−0.24 **	−0.22 **	0.058	−0.35 **	−0.15 **
5. Intrusive rumination	19.70	8.77	0.50 **	0.30 **	0.36 **	0.90 **	1	0.57 **	0.015	−0.008	0.041	0.019	0.012	−0.26 **	−0.26 **	0.048	−0.34 **	−0.17 **
6. Deliberate rumination	18.57	7.20	0.49 **	0.55 **	0.52 **	0.85 **	0.54 **	1	0.029	0.027	0.017	0.038	0.011	−0.16 **	−0.13 **	0.052	−0.27 **	−0.095
7. Social support (total)	23.93	3.60	−0.045	0.078	0.015	0.042	0.022	0.053	1	0.92 **	0.90 **	0.94 **	0.94 **	0.26 **	0.13 **	0.23 **	0.15 **	0.31 **
8. Supervisor support	11.34	2.63	−0.063	0.037	0.010	0.018	−0.005	0.035	0.88 **	1	0.65 **	0.84 **	0.88 **	0.24 **	0.10 **	0.20 **	0.15 **	0.29 **
9. Colleague support	12.60	1.77	0.001	0.10 *	0.015	0.058	0.053	0.053	0.71 **	0.30 **	1	0.87 **	0.81 **	0.22 **	0.12 **	0.20 **	0.10 **	0.27 **
10. Emotional support	11.91	1.96	−0.037	0.10 *	0.045	0.049	0.019	0.059	0.93 **	0.84 **	0.64 **	1	0.87 **	0.24 **	0.11 **	0.21 **	0.14 **	0.29 **
11. Professional support	12.02	1.90	−0.047	0.040	−0.019	0.030	0.022	0.040	0.93 **	0.80 **	0.69 **	0.73 **	1	0.24 **	0.12 **	0.20 **	0.13 **	0.30 **
12. CSE (total)	42.32	5.66	−0.21 **	−0.070	−0.13 *	−0.21 **	−0.22 **	−0.16 **	0.063	0.007	0.12 *	0.028	0.089	1	0.84 **	0.57 **	0.84 **	0.83 **
13. Self-esteem	10.94	1.61	−0.17 **	−0.014	−0.074	−0.15 **	−0.17 **	−0.084	0.035	−0.016	0.092	0.021	0.044	0.80 **	1	0.39 **	0.64 **	0.56 **
14. Self-efficacy	11.33	1.52	−0.042	0.030	0.042	−0.027	−0.020	−0.020	0.058	0.007	0.11 *	0.042	0.066	0.68 **	0.51 **	1	0.19 **	0.34 **
15. Neuroticism	10.18	2.34	0.28 **	-0.20 **	−0.26 **	−0.31 **	−0.28 **	−0.28 **	−0.018	−0.054	0.042	−0.041	0.008	0.81 **	0.57 **	0.34 **	1	0.63 **
16. Locus of control	9.90	1.96	−0.14 **	−0.008	−0.065	−0.13 **	−0.16 **	−0.070	0.11 *	0.089	0.096	0.067	0.14 **	0.75 **	0.43 **	0.37 **	0.44 **	1

* *p* < 0.05; ** *p* < 0.01. SD: Standard deviation, PTSD: Post-traumatic stress disorder, PTG: Posttraumatic growth, CSE: Core-self evaluation.

**Table 2 ijerph-19-11973-t002:** Comparison of mean scores of PTG, PTSD, social support, core self-evaluations, rumination and core-beliefs between French and American firefighters.

	FRANCE	SD	US	SD
**PTG**	50.00	21.74	**56.86**	28.42
**PTSD**	23.69	7.57	24.42	10.14
**Total social support**	23.93	3.59	**28**	3.82
**Emotional support**	11.91	1.96	14.05	2.06
**Professional support**	12.02	1.90	13.92	2.02
**Colleague support**	12.59	1.77	14.30	1.96
**Supervisor support**	11.34	2.63	13.68	2.24
**Core self-evaluations**	42.31	5.66	46.77	6.38
**Self-esteem**	10.94	1.61	**11.88**	1.96
**Self-efficacy**	11.33	1.52	**11.80**	1.69
**Locus of control**	9.90	1.96	**11.60**	1.99
**Neuroticism**	10.18	2.34	**11.51**	2.51
**Intrusive rumination**	19.70	8.77	**21.53**	11
**Deliberate rumination**	18.56	7.20	18.33	8.33
**Core-beliefs**	17.08	6.73	**20.30**	8.20

SD: Standard deviation, PTG: Posttraumatic growth, PTSD: Post-traumatic stress disorder.

## Data Availability

Links to publicly archived datasets analyzed or generated during the study are not available. Data was stored in a private Excel document.

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
