# Peer review of "PTSD and PTG in French and American Firefighters: A Comparative Study"

_ijerph, 2022, doi:10.3390/ijerph191911973_

Round 1
Reviewer 1 Report
Henson et. al., report a comparative study to understand the nature of the relationship between PTSD and PTG from two culturally different population of the firefighters- American and French. Authors found that the PTG is positively related to PTSD and subjective perceptions and the extent of the stress during the event. However, they did not find any substantial relation with social support, core-self evaluations, type of trauma, and socio-demographic variables. Authors reported few other interesting findings such as time since the event and the number of years on the job only predicted PTG in the American sample, while colleague and emotional support only predicted PTG in the French sample. Additionally, American firefighters reported more growth, more social support, more positive self-perceptions, more intrusive rumination, and more neuroticism than French firefighters.
Authors have provided a thorough background literature and strong justification for their hypotheses. They conducted appropriate analyses, and discussed the finding with appropriate explanations. Authors have cited suitable references through the manuscript.
Overall, the current manuscript is excellent and has potential to make the very needed contribution to the field as authors tried to clarify some of the inconsistencies in the literature regarding the mechanisms of post-traumatic stress and growth in first responders and the robustness of the process of PTG. Findings from this study could certainly help future research and as well as help illustrate ways of developing and providing culturally appropriate clinical services for different groups of first responders.
I do not have any major revisions to suggest, however, I do have a few minor concerns, none of them is critical, but I honestly think that by addressing them the article could be substantially improved.
1. One of my concerns is insufficient presentation of the results in form of adequate and separate tables for such an extensive comparative study, while major portion being just summarized in the form of two tables. Overall, it could prove challenging for readers to follow through the results and discussion. For example, the details of responses regarding the post-traumatic events and timing since the events, are not clear and only mentioned briefly in the results and conclusion section despite the fact that aspects of the events could have profound effect on the PTG. Categorical listing of the relevant findings along with the analysis could help readers and substantially improve the overall quality of the manuscript.
2. Additionally, tables could have been used to report the statistical findings from the sections 3.2-3.6 to avoid long descriptive paragraphs, thus making it easier for the readers to refer back and forth between the findings and the discussions.
3. Did authors need to use any approach to minimize bias arising from the linguistic differences of the two groups mainly when screening for the potentially traumatic events? It is possible in such cross-cultural study to have biases with translation as nuances in the statements could be missed that could have led to different scoring, as well as the bias arising from the perception of the raters from the two different countries.
4. Although the explanation for the observation about the gender differences are adequate especially with regards to the firefighters, I believe that the smaller proportion of female responders could potentially contribute to the non-significant finding when compared to the cited studies which have shown a higher PTG scores in women than men.
5. Finally, a self-report instrument about the PTSD and PTG may not be always highly accurate and could represent a perception of PTG rather than an actual psychological change. Do authors believe that combining the ratings from professional clinician for such professionals which they must be going through on timely basis, could help understand the mechanisms and relationship between PTSD and PTG better than the examinations using only a self-report-based scoring?
Author Response
Dear reviewer,
I thank you for taking the time to read and comment my article, and for providing me with such helpful comments. Right below, I have tried to answer each and every one of your concerns.
- I sincerely apologize about including paragraphs relating to the “type of event” and PTG. I had meant to delete those sections, as we considered that our categorization of events was way too wobbly and unprecise. I have deleted those sections, so no further explanations or tables are required in regards to this matter. Additionally, we believe our section on the relationship between time and PTG is sufficiently explanatory and didn’t require an extra table.
- As table 1 sums up all the correlations between PTG and our variables under study, we do not see what extra tables we could add in this paper. However, I did add “see table 1.” in all sections (from 3.2 to 3.6) to help readers refer back to our correlations table.
- Once more, I sincerely apologize about including paragraphs relating to the “type of event” and PTG! I have deleted those sections, so no further explanations are required.
- I completely agree with your comment on gender. The fact that our female portion was (unfortunately) way too small could be responsible for the lack of relationship between PTG and gender. I have added a sentence to point out this limitation in the “Discussion” section of our article.
- You make a great point here. We hadn’t thought about combining self-report measures with the evaluation of a psychologist. This could indeed greatly improve the quality and accuracy of future data. I have added a paragraph about this at the end of the “Limitations” section of the article.
Thank you again for helping me improve my article. I truly hope you find satisfaction in the preceding responses.
Sincerely,
Charlotte Henson

Reviewer 2 Report
Dear authors,
I would like to congratulate you most sincerely on your scientific article, as I consider it to be of high quality and relevance.
I would like to give you some hints that may help you to improve your article:
It is important not to use abbreviations in either the title or the abstract.
The introduction does not explain what kind of work firefighters do in France and the USA. In some European countries they are only responsible for fire fighting.
Hofstede's dimensional model of culture does not seem appropriate next to the objective. I think that before stating the objective, this model could be mentioned and then the objective could be defined for clarity.
In the methodology, the measurement scales used for each variable are adequately explained, but it is surprising that no data on their validity are given. I recommend including them.
Also missing from the methodology is a description of the statistical methods employed and the statistical software used. Furthermore, the ethical aspects of the research could be included in this section, such as the approach to the participants and how informed consent was obtained.
The order of the bibliographic citations is not appropriate and should be modified, following a 1,2,3 correlation, etc., according to their order of appearance.
Something similar occurs with the tables, as table 2 appears before table 1. This aspect should be corrected.
In the tables, everything that appears in abbreviated form should be specified in the form of a legend following the table.
The conclusions are more of a discussion than conclusions. They refer to hypotheses put forward, but there is no hypothesis in the study, only a general objective.
In the conclusions section, all explanatory hypotheses should be accompanied by supporting scientific literature. For example, with regard to the separation of personal and working life in the case of firefighters, favoured by the use of a uniform, no bibliographical citations are provided. The same applies to personality, where explanatory hypotheses are put forward that are not supported by scientific literature. In general, this is the case whenever statements are made about the characteristics of French and American society. Perhaps it would be good to include a section on both in the introduction. Again, in social support, hypotheses are put forward without documentary support.
Why is it concluded that growth development is robust irrespective of cultural group? This statement does not seem entirely consistent, as there are a large number of exceptions and only two cultures have been compared, both of which are Western.
It is recommended to include some limitation of the study.
Nothing more.
Best regards.
The reviewer.
Author Response
Dear reviewer,
I thank you for taking the time to read and comment my article, and for providing me with such helpful comments. Right below, I have tried to answer each and every one of your concerns.
- The literature on Posttraumatic growth shows that a majority of articles use the abbreviated form “PTG” in their title and abstract. However, I do not mind replacing the abbreviation with the full word. I will leave the decision to the editors of the journal.
- I completely agree on the fact that we omitted to mention any potential differences between French and American firefighters’ activities. I consider this a limitation to our study, and have pointed it out in our (newly added) “Limitations” section at the end of our paper.
- I have changed the order of the paragraph and mentioned Hofstede's dimensional model of culture before the “objectives” of this study, as you suggested.
- Regarding the methodology, you ask for additional information about the validity of each used scale. However, the references we give for each scale provides this additional information. For example, the reference given for the PCL-C (“Blevins, C. A., Weathers, F. W., Davis, M. T., Witte, T. K., Domino, J. L. (2015). The Posttraumatic Stress Disorder Checklist for DSM‐5 (PCL‐5): Development and initial psychometric evaluation”) provides information on the internal consistency (α = .94), test-retest reliability (r = .82), and convergent (rs = .74 to .85) and discriminant (rs = .31 to .60) validity of the PCL-C. Thus, we do not consider it indispensable to add this information in the text, as it will lengthen the methods section.
- I have added a description of the statistical methods used in this study in our “Methods” section avec our list of Measures.
- I have added a paragraph on the way we obtained informed consent from our participants in the “Method” section of our article.
- I have changed the citations in text to numbers in brackets, as requested, following a 1, 2, 3 order according to their order of appearance.
- Table 1 now appears before Table 2.
- Abbreviated words are now specified under each table.
- I agree on the fact that the conclusion is more of a “discussion”, which is why I have changed the title of the section to “Discussion”. Additionally, some references to justify some of the ideas mentioned in the discussion were indeed missing. I have added 5 references to better support my hypotheses, especially in regards to the role of the uniform (Pratt & Rafaeli, 1997), the role of subjectivity in the experience of trauma (Weinberg & Gil, 2016), the impact of positive self-view on PTSD (Kashdan, Uswatte, Steger & Julian, 2006), toxic masculinity (Mahalik, Burns & Syzdek, 2007) and societal norms towards men (Sagar-Ouriaghli, Godfrey, Bridge, Meade & Brown, 2019).
- I completely agree on the fact that future studies are necessary in order to confirm 100% the robustness of the process of growth among firefighters, regardless of cultural background. I have added a paragraph on the matter at the end of our “Limitations” section.
- We have added some limitations at the end of our article, in our newly added “Limitations” section.
Thank you again for helping me improve my article. I truly hope you find satisfaction in the preceding responses.
Sincerely,
Charlotte Henson
